# The Social License to Restore—Perspectives on Community Involvement in Indonesian Peatland Restoration

Benjamin John Wiesner * and Paul Dargusch 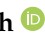

School of Earth and Environmental Sciences, University of Queensland, Brisbane 4072, Australia;
p.dargusch@uq.edu.au
* Correspondence: b.wiesner@uqconnect.edu.au or benjaminjwiesner@gmail.com

**Abstract:** The tropical peatlands of Indonesia are widely recognized as a globally significant carbon stock and an important provider of crucial ecosystem services. However, in recent years they have been increasingly degraded. The Indonesian government has attempted to involve communities in peatland restoration efforts. These attempts were made in recognition of (1) the important role livelihood activities play in land degradation processes and (2) the 'gatekeeping' and stewardship role local communities play in ensuring the durability and longer-term effectiveness of restoration activities. Engaging communities has proven challenging for many reasons, but particularly because of the historical distrust local communities have towards land management interventions. In this article, we borrow the concept of a social license to operate (SLO) from the business management literature to understand why and how community involvement impacts peatland restoration in Indonesia. We introduce the concept and conceptual models of a social license to restore (SLR). As a result of engaging with our perspective, readers will be able to identify how issues of government distrust, low levels of community participation, and poverty—and the counterfactual—may impact the longer-term success of restoration initiatives and how a social license to restore may expedite progress in restoration. Secondly, discussing and linking the multi-faceted issues of peatland restoration will highlight its relevance within the land, biodiversity and human well-being nexus.

**Keywords:** community; involvement; peatland; restoration; social license to operate

## 1. Introduction

Tropical peatlands have been globally recognized for their significance in providing ecosystem services such as housing biodiversity, regulating climate, purifying water and providing raw materials to human populations [1]. Specifically, they are known as one of the world's most significant terrestrial carbon stores, accounting for approximately 15% of global peatland carbon storage [2]. Indonesia, as a single country, contains the world's largest share of tropical peat carbon, with 65% of it [3]. Despite this, in recent years, Indonesia has experienced rapid land-use transformation due to growth in agriculture, oil palm and timber plantations [3–6].

Land conversion for resource extraction and plantations is one of the most obvious drivers of peatland degradation in Indonesia [7]. Activities such as logging and the construction of plantations reduce canopy cover, decrease humidity and limit water retention in the soil, therefore drying out large areas of peat and subjecting them to forest fires which can last and spread for long periods of time [4,8]. Specifically, interventions undertaken as part of large-scale initiatives such as the Mega Rice Project (MRP) have resulted in a large network of drainage canals, monocultures and cleared land plots, which now contribute to a loss in biodiversity, changes in microclimate and increased carbon emissions [9,10].

However, the inclusion of community labor and livelihoods in such practices has also given rise to more subtle but equally important drivers of degradation. Poverty and the associated traditional farming practices or rural communities now play a significant

role in peatland loss [4,11]. Roughly 70% of household income of communities in the Indonesian region of Kalimantan is derived from forestry and farming, with 57% of said households acting on a mere subsistence basis [12]. In the face of recent rates of land-use change, traditional slash-and-burn farming is becoming increasingly detrimental to adjacent peatland ecosystems [13].

In recent years, the Indonesian peatland restoration agency "Badan Restorasi Gambut" (BRG) and non-government organizations (NGOs) have attempted to initiate joint peatland restoration programs with local communities. The aims included raising awareness of peat degradation, restoring degraded peat and empowering communities to support themselves using alternative income sources [14]. Such engagement strategies have proven difficult partly due to a history of distrust and inequality between the government and communities [15]. Communities continue to exploit peat resources, whilst also slowing restoration efforts by knocking down canal blocks and using drainage canals for transportation [9,16,17]. While peatland restoration is recognized as beneficial to the environment, it is simultaneously regarded as disruptive to livelihood practices [18].

It seems that the various Indonesian government agencies, notably at a national level, lack a 'social license' to restore the area. In business ethics, a social license is a type of social contract between an organization and communities affected by its actions. Recognizing and obtaining the importance of such a contract is argued to significantly improve the legitimacy of an activity affecting third parties. The core of this process is creating mutually beneficial relationships and improving the efficiency of stakeholder participation in a given project [19,20]. This paper, therefore, aims to model an approach to obtain a 'social license to restore' in Indonesia and, specifically, Central Kalimantan—a region that contains Indonesia's second-largest peatland area and has seen an intensive application of government-led restoration initiatives over the last decade [21]. By providing background information on the region's history, the role of communities in peat degradation and applying SLO concepts to this context, we aim to develop a guiding model with the purpose of improving future community engagement and peatland restoration in this area. In the context of the land–biodiversity–human well-being nexus, such a model may contribute to ensuring the well-being of rural Indonesian communities and the environment they live within by addressing both as one holistic issue.

## 2. Materials and Methods

The methodology of this study was divided into three parts. Firstly, an analysis of the historical context of land management in Indonesia was conducted, examining, in particular, the drastic land-use changes following the initiation of the Mega Rice Project (MRP) in 1996. This specific point in time was selected due to the tangible links that remain between the immediate and present-day effects of the MRP [5,22]. Additionally, an analysis of the present-day hydrological regimes, population demographics and land ownership in Central Kalimantan was conducted. This first stage was conducted by examining a range of peer-reviewed literature discussing specifically the role of regional and rural communities in the MRP as well as its lasting effects on them. These studies served to inform potential barriers to functioning community-government relationships.

Part two involved an analysis of existing peer-reviewed business literature, discussing, in particular, the origins of social contracts and agreements in community engagement. It served to identify the definition, benefits and potential structural components which could be applied in a land management context. For this, pioneer models of social licensing processes were analyzed and compared to identify the benefits and components of each model, as well as their applicability in ecological restoration. For this purpose, the evaluation of these models was restricted to their core components and underlying principles, rather than specific implementation methods.

Lastly, the analysis of social contract models was utilized to create an adapted and structured community engagement process that specifically targets the challenges of community-led restoration initiatives in Indonesia identified in the historical context. By

examining components of existing social contract models through the lens of the rural Indonesian context, they were adapted to form core pillars in a three-stage process that recognizes the interdisciplinary nature of peatland restoration. Importantly, this process was created to remain applicable in a number of communities that face the same challenges in Indonesia.

## 3. Results

### 3.1. Historical Context

Fire is commonly accepted among rural communities in Indonesia as the quickest and easiest way to make space for crop cultivation and construction. Early iterations of modern-day slash-and-burn farming have also been deeply embedded in the local culture of communities for a long time [11]. Therefore, land cultivation and clearing by rural communities in Indonesia is nothing new.

Traditional farming practices in Centra Kalimantan were first notably disrupted beginning in 1966 when President Suharto initiated large-scale land reform (Figure 1). Forests were cleared for timber production, plantation development or grass fields and migrant farmers from densely populated Java were relocated to assist in crop cultivation [23]. Indigenous land rights were largely overlooked as thousands of forest-dependent communities were displaced through legislative ambiguity, coercion and violence [22,24]. This led to various conflicts between migrant farmers, state forces and concession holders [23].

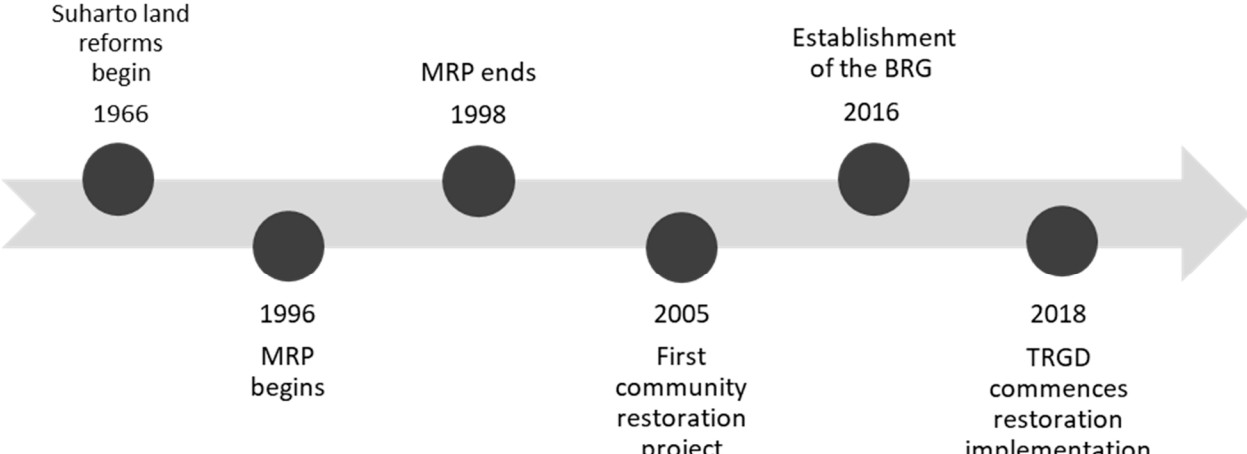

**Figure 1.** Timeline of key land-management events in Central Kalimantan (1966–2018).

The most notable initiative during Suharto's land reform was the "Mega Rice Project" (MRP): a presidential decree which saw one million hectares of peatlands converted into five blocks of rice plantations to support the country's growing food demands [25]. The project involved the construction of a 187 km long canal connecting the region's rivers and nearly 4000 km of additional canals across the project area [26]. Canals succeeded at dropping the water table, allowing for rice cultivation and easy transport of resources. However, acidic peatlands could support no more than a few cycles of rice cultivation, and the project was terminated in 1998 [27].

The MRP left vast areas of peat drained, degraded and prone to fires, which have become especially dangerous when transitioning underground as sub-surface fires cause the most dangerous air pollution [25]. Communities have increasingly contributed to this issue by reducing fallow lengths and increasing clearing and burning to secure remaining land for livelihood activities [11,28]. Slash-and-burn agriculture became the dominant form of land management in post-MRP communities. Rather than traditional swidden agriculture, which involves rotating crop cultivation across small patches of land and allowing for native vegetation to regenerate, slash-and-burn farming involves cutting down and burning vegetation to provide ash-based nutrients for plantation crops [29].

This also involves the replacement of native vegetation with flammable plants such as ferns and sedges around drainage canals [25]. Slash-and-agriculture provides communities with a short-term income from crop cultivation but requires significant land-clearing and increases the risk of fires and pollution. Since the termination of the MRP, various restoration initiatives have therefore taken place in Central Kalimantan.

### 3.2. Restoration Initiatives to Date

First attempts at peatland restoration were made in 2005 by Wetlands International in the former Block A of the ex-mega rice project (EMRP) area, where 20 canal dams were constructed by local communities and contractors using concrete and sand in order to rewet adjacent peatlands [30]. Challenges included misalignment and seepage of materials, but also the destruction of canals by other communities to allow continued use of canals [9,16,17].

Other attempts have since been made by various combinations of academic institutions, government bodies, NGOs and private companies. [18] reviewed four separate restoration projects in Central Kalimantan by various initiators between 2004 and 2016. The projects varied in geographical location, ecological and socio-economic context and had varied goals, including reducing greenhouse gas (GHG) emissions, enabling carbon trading, fostering community livelihoods or promoting environmental education. While challenges among all four restoration projects naturally differed due to their individual context, all projects reported livelihood challenges and negative attitudes of communities towards their restoration project. Specifically, communities found restoration initiatives such as canal blocking to be limiting their livelihood options, which sometimes led to canal destruction [18]. This is consistent with findings from earlier restoration projects [16]. One of the four projects failed to account adequately for the complex demands of communities in their project plan, leading to a lack of community acceptance. Another reported a lack of acknowledgment by the initiators of local community practices, as fire use was banned while having been a part of traditional community practices for generations.

While findings by [18] target mostly private or NGO-led restoration projects, the Indonesian government has since established a centralized government body precisely for this purpose. The peatland restoration agency BRG was established in 2016 by presidential decree, with the task of restoring 2 million ha of peat within 5 years [31]. The agency has set about this task through the RRR Approach: Rewetting, Revegetating and Revitalization of livelihoods. A regional peatland restoration team (Tim Restorasi Gambut Daerah [TRGD]) was established in 2017 and has been actively assisting the restoration implementation since May 2018 under BRG authority [21]. Under its supervision, the agency has supervised the installation of rewetting infrastructure and coordination with academic institutions.

While community perception of restoration is generally favorable, complaints are being voiced of inadequate consultation, disagreements on hydrological management and a lack of financial compensation [21]. Specifically, often only parts of communities are notified or asked to be involved in restoration, which does not achieve free prior and informed consent (FPIC). Negotiations over hydrological management tend to lead to disagreements as the community and government lack understanding of each other's goals. Lastly, financial rewards are needed to incentivize involvement, but such financial resources are often provided to communities negatively affected by restoration, rather than those contributing to it.

Overall, communities tend to participate more in restoration if they receive social support, are more educated, earn a higher income and have more control over the management of their land [32]. Challenges in peatland restoration projects to date reliably point toward a lack of regard for community livelihoods, their traditional land-management practices and general consultation. Historical mismanagement of natural resources by the government fuels disagreements on land management, and communities are not given adequate control over the land they live on.

It seems as if community livelihoods are treated as a barrier to biophysical restoration, rather than a foundation for it. This calls for a more rigorous approach to community involvement in peatland restoration. However, the present-day social and environmental context of Central Kalimantan requires consideration to inform this change.

### 3.3. Present-Day Central Kalimantan

#### 3.3.1. Peatlands and Hydrological Regimes

Tropical peat consists mostly of water with organic carbon content in excess of 18% [31]. Peat swamps are dome-shaped, usually close to fresh-water streams and serve an important function of storing excess rainwater [33]. Under natural conditions, peatlands in central Kalimantan are rain-fed and waterlogged throughout the year. In undrained peatlands, water moves during the wet season, with 90% of rainwater moving through the top layer in a sheet- rather than channel-flow [30]. Dry seasons between April and October bring a drop in the water table, which increases subsidence and $CO_2$ emissions [34]. The decreased soil moisture content also increases fire risk. This is of particular concern during El Niño periods [31]. During the dry season, the water table in even undisturbed peat forests falls in excess of 1 m below ground-level [30].

Canal development has caused a drop in groundwater levels across areas affected by the MRP [35]. Drought is a natural phenomenon and occurs both in the groundwater and soil moisture. However, canalization has amplified drought severity significantly, specifically in the groundwater. Vegetation clearing has been associated with a 2.5× amplification of drought severity both in groundwater and soil moisture [35]. Soil characteristics seem to affect water flow in tropical peatlands. For example, coarse-textured peat has been shown to be at an increased risk of fire if it is drained below a depth of 0.40 m, with 0.85 m resulting in maximum fire risk [36]. Peat humification seems to also affect changes in groundwater level, with more humified peat water tables falling and rising more quickly [31].

A national database for the physical properties of Indonesian peat soils is currently still lacking [31]. However, the authors have suggested that data collection should not be an isolated practice but an on-going monitoring activity that could be part of a restoration initiative. In the absence of a national database, peatland restoration projects need to account for the local physical soil properties through individual monitoring programs.

#### 3.3.2. The Population and Land Tenure of Central Kalimantan

The EMRP area is home to a diverse range of transmigrants relocated during the MRP as well as long-term Dayak residents, including Ngaju, Ot Danum and Maanyan people. These groups traditionally practiced hunting, fishing and swidden agriculture, but have increasingly adopted slash-and-burn farming to quickly source a short-term income [6].

The Central Kalimantan province contains 14 regencies and roughly 1569 villages with a total population of roughly 2.5 million people [37]. Mapping by [38] has shown that roughly half of all land in central Kalimantan is dedicated to production forests and convertible production forests. While this land is dedicated to agricultural production or resource extraction of some kind, large parts of it are not covered by any kind of license and restoration activities are both possible and even prioritized in many production areas [39]. Plantations cover 11.34% of the region's landmass, while protected forests and nature reserves take up roughly 10% and 2%, respectively. Settlements occupy 8% of central Kalimantan, but communities are active across production forests and settlements alike.

Land rights in Central Kalimantan are communal and depend on the location, cultural values and labor input [40]. Unlike land occupied by plantations, community land rights often overlap or clash, for example, with investor rights to bid for the land. Only transmigrants have historically been granted formal land titles when they were relocated [41]. Dayak people have been residing in Central Kalimantan for a long time but often cannot obtain formal land titles, instead relying on unregistered documents which are difficult to register due to financial or legal hurdles. Considering these barriers, villagers often accept "compensation" from investors seeking to establish plantations, as they require the

financial resources to support their livelihoods [41]. A lack of formalized community land rights is, therefore, a significant barrier to long-term income generation for communities, which precludes them from effectively engaging in restoration activities.

### 3.4. Key Identified Challenges

An analysis of the historical and present-day context of land management in the Central Kalimantan province shows that communities face a number of socio-economic, legal, financial and environmental pressures, which has led to challenges in effectively involving them in peatland restoration. These challenges can be categorized into government distrust, superficial mechanisms for involvement in restoration activities and poverty among the communities involved [12,15,42,43].

#### 3.4.1. Lack of Governmental Legitimacy

Past marginalization of communities and the ambiguity of land rights have sparked distrust of communities towards the government. Farmers and rural inhabitants are generally not opposed to restoring forests and switching to more sustainable farming practices [44]. However, the Indonesian government lacks legitimacy as the effects of past land reform attempts during the MRP are still felt by communities [42]. A present-day lack of formalized land rights exacerbates this, as communities cannot predict or control the future of the natural resources they depend on [31]. The lack of trust both in the interests of the government and methods employed for restoration threatens restoration success.

#### 3.4.2. Instrumental Participation

Current restoration measures are being questioned by affected parties as they do not always account for the effect of restoration on livelihoods [45]. Communities are involved in on-ground restoration but not the decision-making process, as they are often not consulted, or their views are not adequately reflected in restoration plans [21]. This reflects what has been described as "instrumental participation" [43,46]. This is undesirable as it removes a sense of ownership and place, which mirrors the challenges faced by communities during the MRP [40]. The nature of participation also impedes the level of understanding people have of the project, its means and its necessity. This represents a lack of transparency, which sparks further distrust and misunderstanding. For example, fire use is currently banned in many areas even though it is socially and culturally ingrained in the way of life of many communities. Hence, it is still practiced in secrecy by individuals who then have less control over it [13].

#### 3.4.3. Poverty

Poverty is widespread among many rural peatland communities in Central Kalimantan [12]. As a result, people source income with a short-term perspective regardless of environmental consequences [47]. The changes to shorter fallow periods occur due to a lack of short-term income alternatives [48]. Similarly, communities accept compensation from plantation investors, giving away the little land rights they have for money. The alternatives often follow a long-term strategy and require control over resource distribution, which is problematic considering the lack of spare time when the focus lies on short-term survival. Poverty is, therefore, currently impeding the efficiency of the project due to dependency on short-term income and an associated lack of time.

### 3.5. Social License to Operate Concepts

The three overarching issues communities face in rural Kalimantan can be argued to preclude the Indonesian government from gaining their approval. In business literature, such approval has often been termed a "social license to operate" (SLO). The concept of an SLO is an extension of how firms seek legitimacy for their actions through corporate social responsibility initiatives [49,50]. It is also based on contractarianism and follows the basic principles of a social contract [51]. The need for such a contract historically developed

out of the recognition that a competitive market is not always mutually beneficial [52]. Along with gradually changing attitudes towards industries affecting the environment, this created a need for regulation, specifically within the context of adjacent communities that are directly affected [53]. The SLO, therefore, developed as a license sought after primarily by resource extraction companies in pursuit of legitimizing their actions [54].

This benefits these companies as the elements of consultation and transparency lead to higher community acceptance [55]. A lack of such acceptance causes delays and costs to a given project [56].

A social license to operate represents a community's approval of a third party and its operations in the local environment [19,57,58]. Some scholars distinguish further between a normative and strategic SLO, the former representing legitimate justification in the eyes of all stakeholders [19]. The latter often represents gaining the approval of the most influential stakeholders, making it easier to obtain [50].

### 3.5.1. Components of a Social License

A company's success in obtaining a social license depends on its past interactions with the stakeholders it is aiming to obtain this license from [59,60]. This has been commonly termed moral legitimacy, a factor that influences the SLO process but is also influenced by it [61]. Moral legitimacy may, therefore, simply be referred to as the reputation of an entity seeking an SLO.

Given the reputation, scholars have furthermore identified legitimacy, trust and consent as three influential factors in shaping an SLO (Figure 2) [51,57,62]. As part of this, a further subdivision into subcategories for trust and legitimacy was proposed [57].

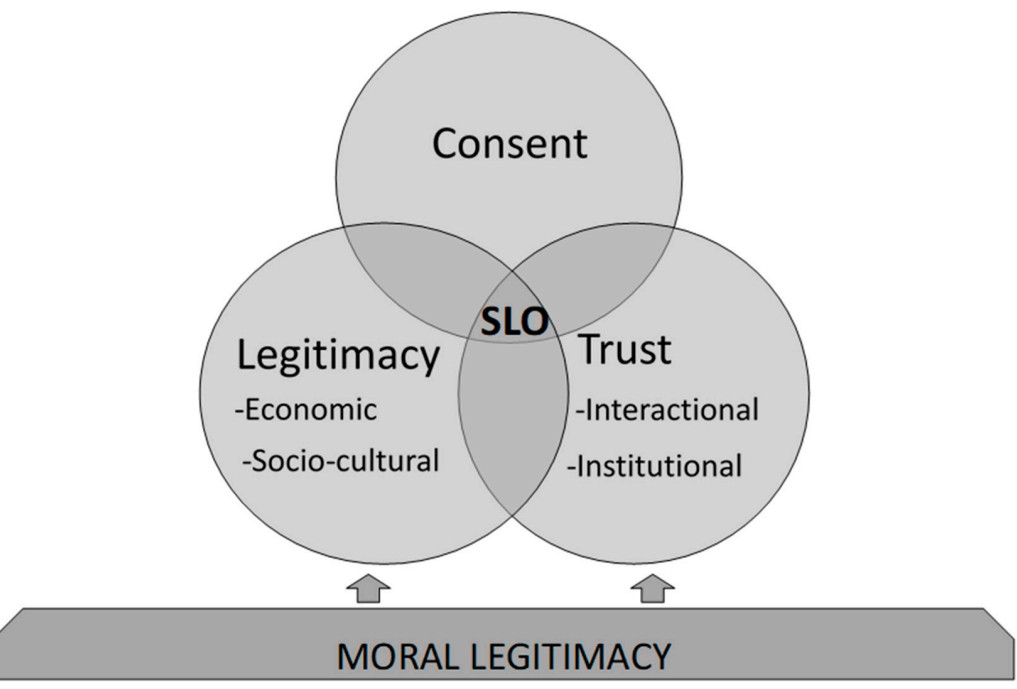

**Figure 2.** Model representing the academic consensus on factors influencing the development of a social license to operate (SLO).

- Legitimacy

Project legitimacy differs from moral legitimacy (or reputation) in that it refers to the project at hand, rather than past events or interactions of the entity. Within this context, the legitimacy of a project can be divided into economic and socio-cultural legitimacy [57]. Economic legitimacy represents the community benefit, which may represent higher incomes or employment opportunities. Community knowledge factors into this, as a community might not be fully aware of the potential impact of a project, rendering it vulnerable to mis-

information [63]. Furthermore, perceived economic legitimacy will be affected by feedback, as a social license is continuously re-evaluated. Aesthetic changes might affect community perception, for example, through the visual appearance of new infrastructure [64].

Socio-political legitimacy refers to the degree to which a project accounts for the local culture and societal boundaries [57]. This is important in the face of a possible 'excess of objectivity', which can lead companies to develop generalized solutions while disregarding social issues in the process [65]. Secondly, it limits the risk of 'checking the box' of community approval rather than aiming for a mutually beneficial relationship [49]. In order to obtain socio-political legitimacy, some argued for risk mitigation and remediation of potential harm as well as transparency throughout the project [51]. Within the socio-political context, this would mean showing awareness of the local context and engaging the community in decision making. This ensures mutual understanding and risk mitigation, although sacrificing some decision-making power might affect the desired outcome.

- Trust

Trust is referred to as the belief of one stakeholder that another will match their behavior to the given expectations [55]. Interactional trust and institutional trust have been recognized as important contributing factors to an SLO [57]. Interactional trust largely refers to the quality and quantity of interactions between parties involved in an activity. However, some scholars also include fairness in the procedure [55]. Although transparency and inclusivity contribute to this, it is important to consider barriers to certain types of knowledge that might be lacking in a stakeholder relationship [66]. Furthermore, distributional fairness contributes to trust by making sure all parties are satisfied with their share of any benefits [58].

Institutional trust is referred to by as psychological identification between stakeholders through a continued investment in each other's aims [57]. It is also a general view of each other outside of everyday interaction [51]. Interactional factors over a period of time can affect this general view since a community might start trusting representatives they interact with. However, the company might also empower the community to judge it based on discursive and practical knowledge (e.g., through education). Institutional trust can then develop as part of a positive feedback loop similar to economic legitimacy. Conversely, a single catastrophic event can destroy a government's reputation, similar to the events following the MRP.

- Consent

Consent for an activity is present when all stakeholders affected by it approve of the activity without being influenced through coercion or misinformation [51]. However, in the context of an SLO, it cannot be referred to as only 'prior', as an SLO is continuously re-evaluated, and consent must be maintained, rather than obtained only once [59].

3.5.2. Applying SLO in Indonesian Government-Led Restoration

These components have found application mainly within private profit-oriented enterprises. However, we argue that it can find application in government-led peatland restoration initiatives for three main reasons:

Firstly, the fundamental scenario in both business and not-for-profit cases is the same. Communities are affected by adverse impacts of land-management initiatives of a larger cooperation or agency, which raises the need for adequate cooperation to achieve a mutually beneficial outcome. While private enterprises and public agencies might have different goals, the SLO process is focused on community interests which remain similar in both cases. In fact, one might argue that SLO finds even better application in a public scenario where outcomes provide long-term benefits to communities rather than short-term economic gain. This is evidenced by past attempts of forestry companies to implement SLO in their community engagement practices in Indonesia. Both forestry companies assessed in a case study by [67] struggled to gain legitimacy with communities due to concerns regarding the impacts of forestry on their long-term livelihood activities. The companies could offer little

beyond financial compensation, which stands in contrast to government-led restoration initiatives which target long-term peatland rehabilitation [18].

Despite these long-term benefits, communities must be able to generate income through the lifespan of a restoration project, which raises the question of whether not-for-profit projects can still generate sufficient income for communities. Successful government-led restoration should enable a stable, long-term income stream for communities, removing the need to exploit peatlands in the short term. In contrast, privately led initiatives have provided short-term income, but no avenues for long-term economic stability [67]. While short-term income is needed to assist communities in transitioning to more sustainable livelihood practices, roughly USD 200 million have been pledged to the initial stages of the BRG restoration initiative to assist this cause [68].

Lastly, lessons from previous restoration cases highlight challenges that closely link to the social license components discussed above. For example, government legitimacy is lacking due to past failed attempts at land reform and communities lacking formalized land rights [41]. Trust seems to have been lacking in past restoration projects, as communities actively disrupted rewetting activities out of a lack of trust in their benefit [16]. While restoration measures may cause negative side-effects, even when correctly implemented, trust is needed to gain community support despite these effects. Lastly, it was noted that consent from communities was not always established, as communities were either not informed or not asked to be involved in restoration [21].

### 3.5.3. The Social License to Restore (SLR)

In light of these reasons, we argue that SLO can find use in a government-led restoration initiative if its components are adapted and implemented as actions rather than components. The SLR model involves three stages, firstly creating a mutual level of understanding, then engaging the community through involvement, interaction and intervention and finally evaluating progress through relevant performance indicators (Figure 3). These stages do not occur in order, but rather as part of one integrated process. Each stage reinforces the other two and relies on them at the same time. This reflects the continuous nature of the SLO process [19].

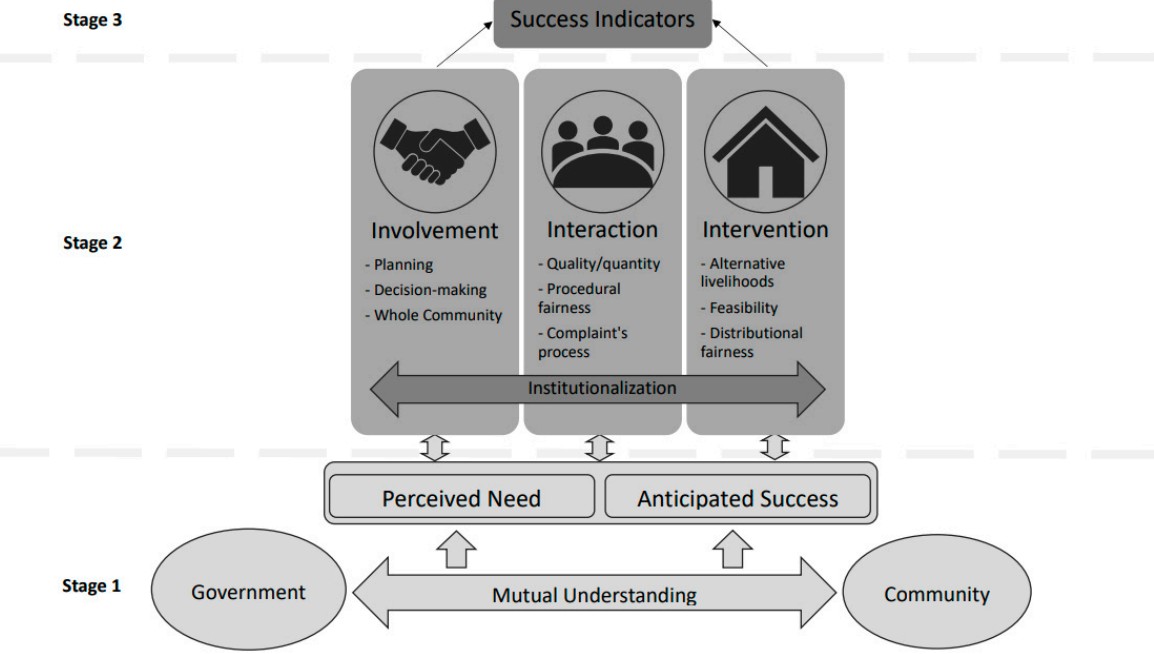

**Figure 3.** Three-stage model for developing a social license to restore (SLR).

- Stage 1: establishing common ground

A social license to restore relies on a mutual understanding and equal footing between government and community as its foundation. However, the relevance of the process does not depend on a perfect agreement between both parties since perception disparities are recognized and targeted in the first stage. In fact, the destruction of dams or the use of canals indicate that both stakeholders do not share the same perception of the employed strategies [16]. Perception disparities represent fundamental points of conflict and can occur due to a low level of education or misinformation in some communities, as well as a lack of awareness by the government of the challenges communities face [45]. This could be improved through education on peatlands, their restoration and economic incentives, hence laying a foundation of economic legitimacy [57]. Methods for this might include schooling, social media or public government banners. However, inherent government distrust must also be tackled through direct interaction, for example, between government representatives and communities. Ideally, stage 1 fosters anticipated economic benefit resulting from the project as well as the perceived need for a change in environmental management, hence adding environmental to economic legitimacy. This lays the groundwork for overcoming a lack of trust (or moral legitimacy) for the purpose of restoration. However, anticipation also relies much on the project's execution, which links this stage to the second.

- Stage 2: pillars of participation

The proposed model is largely centered around stage two and its three main pillars: involvement, interaction and intervention. This stage is strongly linked to the other two, as a restoration project can only be implemented when communities perceive a need for it and its potential benefits. However, this perception also relies on reinforcement through performance indicators (stage 3). The three pillars are, therefore, naturally placed in the middle.

1. Involvement

The involvement pillar represents one side of a relationship which gives both parties an equal standing. It requires the government to involve communities in the planning and decision making of a restoration project depending on local needs and preferences. To make this possible, it builds on preliminary education to give communities a role above just providing labor. Involving communities in the planning of restoration activities provides them with an understanding of the background and reasons for said activities. It also further informs them of long-term economic benefits, which are a high priority for improving community involvement [69]. Next, involving communities in the decision-making process provides a basis for ensuring socio-cultural legitimacy, as it more easily raises awareness of socio-cultural implications of certain activities, such as a fire ban [13]. In this context, it is not always necessary or beneficial to give full decision making control to communities. Rather, a consulting role can provide empowerment, responsibility and intellectual involvement, which acts as a countermeasure to current government distrust [70]. Instrumental participation is therefore elevated beyond labor to become an actual intellectual contribution and transparency. In this process, it is important to include the entire community in one way or another, for example, by identifying and communicating through influential individuals in a village [71]. This helps avoid discrepancies in knowledge or involvement, leading to a potential divide.

2. Interaction

Interaction is the government's main resource to improve its reputation or moral legitimacy with communities. Direct interaction with communities is crucial to avoiding conflict [12]. Current government distrust can be partly overcome through involvement. However, high-quality and -quantity interactions between the government and communities are crucial to establishing a stable relationship [55]. This could be achieved by assigning facilitators to each village to uphold communication, since trust is more easily developed towards a single person than an entity or corporate body [51]. Communication should include the regular exchange of information, best-practice methods and demonstrations

of successful pilot projects for motivation. Policy changes and any background to such changes should also be communicated. This serves as an extension to the involvement tool, but beyond that, also as a grievance or complaints process. It shows that potential issues are considered first, and that communities are listened to, should problems arise. For example, a concern regarding unequal distribution of workload or economic benefits could be raised with a facilitator. This reflects the value of procedural fairness in building trust as well as socio-cultural legitimacy [55,57].

3. Intervention

The intervention pillar represents the counterpart to involvement by giving communities a space to communicate their challenges in generating income and managing their land sustainably. This makes it possible to jointly develop ways of overcoming these barriers as a foundation of successful community involvement. Importantly, the intervention phase must be initiated by the government while giving communities full freedom to voice concerns and suggest solutions, which is enabled in the involvement phase. However, progress in government-led peatland restoration has been limited mainly by socio-economic challenges faced by communities [18]. Successful intervention will need to involve understanding community concerns and optimal short- and medium-term income solutions to aid the transition to sustainable income generation. This could be achieved through regular open forums with communities and government representatives, held consistently before project invitation until monitoring after completion. Concerns could also be anonymously voiced through key community representatives. Intervention might also involve negotiation on more formalized land rights, which enable communities to pursue more long-term livelihood options. The intervention pillar builds on the previous two since local knowledge is crucial to developing locally feasible livelihood options [71]. Distributional fairness is important to ensure an even spread of benefits to avoid socio-cultural issues [58]. Intervention links back to reinforce the economic legitimacy of the restoration process since it ensures activities can be carried out without incurring a loss in livelihood.

The three tools act in accordance with each other. Through time, if used correctly, they can result in the institutionalization of the relationship between government and community. This reflects a regard for each other's interests as well as matching values. Whilst the lack of current moral legitimacy of the government will not be overcome through a single successful initiative, psychological identification in the context of one project can help alleviate distrust.

- Stage 3: success indicators

The last stage serves to verify both previous stages by providing indicators of the perceived legitimacy, trust and consent within the project. Such indicators aid in gauging community approval, their awareness of project aims and the effectiveness of livelihood alternatives. For example, community awareness might be evaluated through regular interviews and discussion forums. Livelihood options might be compared through changes in income or resource availability, which respectively measure economic benefit and sustainability. Communities should be aware of the importance of indicators and partly in charge of developing them. Such indicators can then serve the community to empower them—to evaluate their own progress and that of the government. Additionally, they may inform necessary changes to livelihood alternatives, available resources or strategies to address any identified shortcomings within the project. Results reaffirm the perceived need and anticipated success in the best of cases. Otherwise, they provide information to improve the project. Lastly, indicators of successful integration of the SLR models should link to biophysical indicators of restoration progress through education. For example, community awareness of changes in water tables, peat depth or soil consistency will link to the related condition of community livelihoods. In an ideal case, communities become champions of their own livelihoods but also stewards of the environment they live within,

contributing to a much-needed national understanding of the biophysical condition of peatlands [31].

### 3.5.4. The Novelty of SLR

The idea of holistic perspectives on human-environment relationships and the importance of community engagement within this space is not new. Concepts such as socio-ecological systems (SES) or coupled human-environment systems (CHANS) have described the inherent interconnectedness of social, ecological, economic and political components of human-environment relationships [72,73]. Interactions between these components are argued to be increasing in scale and pace, highlighting the need for an improved understanding of their extent [74]. SLR adopts this perspective as its foundation and provides specific actions to create a more sustainable relationship between Indonesian communities and the peatlands they occupy. The concept is new to Indonesia and presently confused with similar concepts such as corporate social responsibility (CSR) or FPIC [67]. However, SLR exceeds these approaches by not merely seeking community approval or tolerance but building on it to achieve genuine investment in a mutually beneficial outcome. In the Indonesian context, rather than seeing community livelihoods as a problem to address, it develops them as a foundation of successful biophysical rehabilitation. By establishing trust in an initiative and improving government legitimacy through continued interaction, community engagement can cease to be transactional and instead become genuinely effective.

### 4. Conclusions

The peatlands of Central Kalimantan are threatened by communities depending on short-term income gained from resource extraction and slash-and-burn farming. A history of distrust and inequality, lack of land rights and community poverty has negatively affected attempts of the government to engage communities and initiate peatland restoration in adjacent areas. Termed moral legitimacy in business literature, the government has an unfavorable reputation among local communities. The social license to restore adapts social license concepts to establish not just community involvement, but trust in positive restoration outcomes. It achieves this by establishing an equal relationship between communities and their government which enables effective communication, bringing about change in both environmental conditions and livelihood measures. The model employs the three pillars of involvement, interaction and intervention to:

- Elevate participation through increasing transparency, ownership and awareness of communities in the project;
- Build trust and socio-cultural legitimacy through frequent interaction, relevant success indicators and community involvement in biophysical monitoring;
- Establish sustainable, long-term income through identifying feasible livelihood alternatives and exploring formalized land ownership options.

The three main tools act in support of each other, over time overcoming the noise of historical distrust and forming an institutionalized relationship. Success indicators aid in evaluating and improving the procedure of the project and encourage continued involvement by reinforcing the need for it.

Implementing the SLR model may be impacted by residual inequalities in the power dynamics within land management in Indonesia. The continuous nature of the SLR process also requires a level of stability that might be impacted by changes in government in the long term. Future research might aim to trial the SLR model using short- to medium-term success indicators and linking them through overarching long-term goals. Comparative studies between communities implementing SLR and conventional methods could shed light on socio-economic and biophysical benefits of the concept. Importantly, the SLR process is not an independent guide to community-run restoration. Rather, it serves as an extension to current approaches by improving community approval of restoration communities. In any case, the high resource requirements of large-scale peatland restoration call for increased autonomy of communities, enabling self-management of their livelihoods,

environment and biodiversity in the long term. This can be supported in the early stages by creating communication networks between communities, giving them more leverage and enabling mutual learning, shared resources and long-term cooperation.

**Author Contributions:** P.D. developed the idea of a restoration-specific social licensing process; B.J.W. conducted preliminary and background research and developed the model/process for the social license to restore. All authors have read and agreed to the published version of the manuscript.

**Funding:** This research received no external funding.

**Institutional Review Board Statement:** Not applicable.

**Informed Consent Statement:** Not applicable.

**Data Availability Statement:** Not applicable.

**Conflicts of Interest:** The authors have declared no conflict of interest.

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
