# Peer review of "The Social License to Restore—Perspectives on Community Involvement in Indonesian Peatland Restoration"

_land, doi:10.3390/land11071038_

Round 1

Reviewer 1 Report

The paper presents an interesting idea that is clearly described, using a writing style that is very easy to read. In my opinion, the proposal it presents can be of great interest for scientists and practitioners of restoration alike. I detected some typos (which I highlighted in yellow in the attached PDF file) and a few minor issues that I highlighted in the text and further explained in pop-up comments in the PDF file. All in all, I recommend this paper for publication after these minor issues have been addressed.

Author Response

We thank you for your feedback and have made changes appropriate to your comments, all of which can also be found in the table below:

No.

Comment/Recommendation

Old position

New Position

Response

1

I detected some typos (which I highlighted in yellow in the attached PDF file)

All highlighted spelling errors have been corrected.

2

I feel like this needs an introduction, since not every reader is familiar with this noun. I suggest something like “The Indonesian region of central Kalimantan”.

Line 45

Line 51

This sentence has been rephrased to “Roughly 70% of household income of communities in the Indonesian region of Central Kalimantan (…).”

3

This is not in the reference list

Line 55 and 142

Both references has have been removed in the revised manuscript

4

I think it is necessary to explain why the authors chose to focus on this region

Line 65

Line 81

The manuscript has been adjusted to include the following justification:

"(…) on Central Kalimantan: a region which contains Indonesia’s second largest peatland area and has seen intensive application of government-led restoration initiatives over the last decade (Januar et al. 2021).”

5

I think the authors need to include the date of this event, most readers do not know it.

Line 72

Line 93

Rephrased to “drastic land-use changes following the initiation of the Mega Rice Project (MRP) in 1996.”

6

This appears as of 2014 in the reference list. Please check and correct it.

Line 199

Line 434

The reference year has been corrected to “2014” in-text and in the reference list.

Reviewer 2 Report

No comments.

Author Response

We thank you for your feedback. The introduction has been improved to provide a more well rounded perspective of the article

Reviewer 3 Report

Summary

The aim of the paper is to apply a theory used mainly in business management literature to a natural resource management issue. The social license to operate (SLO) is used to understand why and how community involvement impacts peatland restoration in Indonesia. The paper’s main contributions are the novel application of this theory and the conceptualisation of a new conceptual model, the social license to restore (SLR). It is important to note that this paper is a perspective piece and while academic rigor is still highly important, qualities such as originality and academic significance were given more weight in this review.

This article provides a unique insight into a relevant business management theory and applies it to a topical and important issue (peatland restoration in Indonesia). The introduction of the concept of a social license to restore has implications for wider peatland restoration initiatives that require engagement between governments/businesses and local communities. In this context, the concept is an original contribution to the literature and has the potential to contribute significantly to academic debates around drivers of restoration success.

General comments

The introduction provides a good and concise overview of the unique issues facing efforts to protect tropical peatlands in Indonesia. The relevance of the present study is made clear in the context of the government's inability to engage properly with communities living near peatlands. However, the concept of the 'social license to restore' is introduced with little context or definition. It is unclear to the reader how a social license is obtained and what the relevance of this concept is to the context of peatland protection in Indonesia. Although this is addressed later in the manuscript, a sentence or two on this issue in the introduction would be helpful to prime the reader. Furthermore, it is not immediately clear how this article contributes to a conversation about the land-biodiversity-human wellbeing nexus which is pertinent to this section of Land.

The authors reference the issue of land conversion for resource extraction and plantations as a significant driver of peatland degradation and focus the first part of their methodology on land use change caused by the Mega Rice Project (MRP). While selection of this project fits under the umbrella of land use conversion for plantations, it was not made entirely clear why this driver of peatland degradation was chosen above others, and more justification of this choice would be welcome. Part two of the methodology is sound and provides an original and novel application of a business model to a socio-ecological context. The last part of the methodology integrates the two first parts, culminating in the development of a new conceptual model which makes a useful and unique contribution to the academic literature. However, in this final component of the methodology the authors talk about “an evaluation of underlying values” which is not revisited later in the paper. It would be useful if these values can be clearly defined, as well as the method for evaluating them, or if this can be removed from the methods if this was not done, and if the language of “values” can be used in the results section to signpost where the results of this evaluation are described.

The history of peatland degradation provided useful context for the study. While it was clear why the peatland areas were degraded initially (economic gain) it was less clear what motivated the restoration process. The narrative implies that actions were taken to improve the health of peatlands, however it seems this was done so to secure sustainable economic opportunities rather than environmental protection. Both are valid reasons, however perhaps the authors can make this duality clearer in the text. The reasons for lack of community support are outlined clearly using subheadings and references are provided to substantiate claims, however more explicit use of examples or cross-reference to the context of the MRP might help connect section 3.1.1 to the initial aim of the methodology.

The authors claim in section 3.1.2 that lack of community input has exacerbated socio-cultural issues, however this is not substantiated by a reference. While an example is provided, it is not immediately clear how this demonstrates that there is a socio-cultural issue. This section risks generalising wider theories regarding the benefit of community involvement in decision-making to the Indonesian context. Further research in this section is needed to make the argument that superficial mechanisms for involvement in restoration activities is an issue identified by members of Indonesian communities. 

The concept of an SLO is well explained and it is clear how this concept would be relevant to the context of resource extraction in peatlands. The components of a social license are detailed nicely and in reading through the sections it becomes implicitly clearer how this concept is applicable to the Indonesian context.

The three-stage model is explained clearly however the image quality is quite poor making it difficult to read. The concept of moral legitimacy is not clearly outlined in any stage of the process despite being discussed at length in the previous section. Integration of this concept into the model will help connect the relevancy of the two sections.

Subsequent sections clearly outline the processes in each stage and apply it to the Indonesian context. However, while the authors mention that the model is applied to peatland restoration at a community scale, it is not immediately clear upon reviewing the model that this has been tailored to either a tropical peatland or Indonesian context. The model, as is, seems to work well as a general base, however another rendition of the model with examples of the Indonesian context embedded in the diagram would make its relevance more immediately clear. That said, the sections as mentioned do very clearly outline the authors arguments for the model's relevance. Additionally, the concept of underlying values was discussed in the introduction of methods, but it is not clear which values were identified and whether these are indeed shared by Indonesian communities.

The section on involvement seems to miss a key aspect of ensuring effective co-management and deliberation which is identifying the needs of the community before imposing the needs of the business/government. The tone of this section seems to come from a top-down rather than a bottom-up perspective (e.g. see Reed et al’s “theory of participation” (2018) for a range of different modes of engagement and factors that can explain the success or otherwise of engagement). There is a strong focus on the need for intellectual involvement and education of the community in this process, however not every community member may have the capacity for this type of engagement and yet their values, wants, and needs must be considered. Perhaps more research is needed in this section to demonstrate that the authors understand that community members' values and needs must also (and arguable first) be considered to ensure mutual understanding. Similarly, the section on interaction seems to place more emphasis on the aspect of information dissemination from companies/governments rather than the process of aiming to understand the daily struggles and unique circumstances affecting community members. In the section on intervention, it is not immediately clear who should initiate the interventions or how these statements apply in the Indonesian context. Examples of methods to discuss and implement interventions in the Indonesian context would be helpful. Success indicators in the last stage could include co-development of appropriate strategies rather than simply community approval and awareness. Community awareness could be evaluated through non-invasive measures such as observation of community behaviours and compliance with restoration strategies. Again, it is unclear whether these success indicators consider the unique values and perspectives of Indonesian communities.

While this paper introduces a novel concept and provides some legitimate examples for how the new model applies to the Indonesian context there are some clear limitations which should be outlined in the manuscript. Some of these include 1.) whether the Indonesian government would elect to implement a strategy like this given the historical lack of consideration for local community involvement, 2.) the difficulty in enforcement capacity of this model due to historic and highly engrained power dynamic inequalities, and 3.) lack of clarity regarding the amount of time the process of obtaining an SLO would take and the conditions that must be considered for a successful start to this process. This model also does not account for understanding and preventing illegal resource extraction activities. Considering the aims of the paper, the authors do not adequately explain how this model could enable communities to actively restore peatlands, for example highlighting future research to assess the implementation of the model in restoration efforts. Careful consideration of further limitations of this model can help inform future research and allow others to attempt to implement this model.

Specific comments

  • Lines 60-63: Consider shortening or rewording this sentence to read more clearly as it is currently a run on.
  • Line 101: Can you define the three major area categories?
  • Line 112: Using the term degraded implies negative consequences for biodiversity and ecology, but not necessarily for rice production. It seems as though the communities and national government were motivated to clear more lands and restore degraded peatlands for continued economic production, perhaps this can be made clearer.
  • Line 145: "A lack of community input has exacerbated socio-cultural issues." A reference to the relevance of this statement in the Indonesian context would help qualify it.
  • Line 195-196: Socio-political legitimacy is identified as a form of legitimacy however the figure shows socio-cultural. Perhaps these two should be aligned.

Overall Recommendation

Reconsider after Minor Revisions.

While generally this is a good piece of work and a novel contribution, the lack of consideration for local community perspectives leaves this paper feeling unbalanced and renders some of its recommendations and conclusions potentially insensitive and underdeveloped. Minor revisions should be made before reconsidering this paper.

Alexa Green

Mark Reed

Author Response

We thank you very much for your extensive feedback. We have made major revisions to the manuscript, which also addressed your comments specifically. While you deemed the introduction and background sufficient, it has been expanded due to the request of another reviewer. Please note that due to the tracked changes function, accepting all changes to view the manuscript might result in some slight formatting issues which can be corrected if/when the manuscript is accepted. Please find our responses to your comments in the table below. We look forward to hearing back from you:

No.

Comment/Recommendation

Old position

New Position

Response

1

The concept of the 'social license to restore' is introduced with little context or definition. It is unclear to the reader how a social license is obtained and what the relevance of this concept is to the context of peatland protection in Indonesia. Although this is addressed later in the manuscript, a sentence or two on this issue in the introduction would be helpful to prime the reader.

Line 59

Line 70

The manuscript has been rephrased to provide a brief definition of a social license in the following form:

“In business ethics, a social license is a type of social contract between an organization and communities affected by its actions. Recognizing and obtaining the importance of such a contract is argued to significantly improve the legitimacy of an activity affecting third parties. The core of this process is creating mutually beneficial relationships and improving efficiency of stakeholder participation in a given project (Wilburn & Wilburn 2011; Provasnek et al. 2017).”

2

Furthermore, it is not immediately clear how this article contributes to a conversation about the land-biodiversity-human wellbeing nexus which is pertinent to this section of Land.

Line 22

Line 86

Line 746

A section has been added to the manuscripts abstract, introduction and conclusion to specifically link to the discussion of the land-biodiversity-human wellbeing nexus:

Abstract:

“Secondly, discussing and linking the multi-facetted issues of peatland restoration will highlight its relevance within the land, biodiversity and human well-being nexus”.

Introduction:

“In the context of the land-biodiversity-human well-being nexus, such a model may be pertinent to ensuring the well-being of rural Indonesian communities and the environment they live within by addressing both as one holistic issue.”

Conclusion:

“Importantly, the SLR process is not an independent guide to community-run restoration. Rather, it serves as an extension to current approaches by improving community approval of restoration communities. In any case, the high resource requirements of large-scale peatland restoration call for increased autonomy of communities, enabling self-management of their livelihoods, environment and biodiversity in the long-term.”

3

While selection of this project fits under the umbrella of land use conversion for plantations, it was not made entirely clear why this driver of peatland degradation was chosen above others, and more justification of this choice would be welcome. 

Line 35

Line 37

The manuscript focuses mainly on land-use practices among communities and their contribution to peatland degradation.

Land-use conversion for plantations was introduced in the manuscript for the purpose of painting the bigger picture of peatland degradation and to not ignore its primary drivers. While plantations are not the primary focus of this paper, they gave rise to modern community land-use and therefore provide important background. We recognize that this distinction was not entirely clear and have therefore adjusted the manuscript in the introduction:

“Land conversion for resource extraction and plantations is one of the most obvious drivers of peatland degradation in Indonesia (Abood et al. 2015). Activities such as logging and the construction of plantations reduce canopy cover, decrease humidity and limit water retention in the soil, therefore drying out large areas of peat and subjecting them to forest fires which can last and spread for long periods of time (Dohong et al. 2017; Turetsky et al. 2015). Specifically, intervention undertaken as part of large-scale initiatives such as the “Mega Rice Project” (MRP) have resulted in a large network of drainage canals, monocultures and cleared land plots, contributing to biodiversity loss, changes in microclimates and increased carbon emissions However, inclusion of community labour and livelihoods in such practices has also given rise to more subtle but equally important drivers of degradation.”

4

However, in this final component of the methodology the authors talk about “an evaluation of underlying values” which is not revisited later in the paper. It would be useful if these values can be clearly defined, as well as the method for evaluating them, or if this can be removed from the methods if this was not done, and if the language of “values” can be used in the results section to signpost where the results of this evaluation are described.

Line 87

Line 112

The manuscript in this section of the methodology has been rephrased:

“By examining components of existing social contract models through the lens of the rural Indonesian context, they were adapted to form core pillars in a three-stage process which recognizes the interdisciplinary nature of peatland restoration. Importantly, this process was created to remain applicable in a number of communities which face the same challenges in Indonesia.”

5

The narrative implies that actions were taken to improve the health of peatlands, however it seems this was done so to secure sustainable economic opportunities rather than environmental protection. Both are valid reasons, however perhaps the authors can make this duality clearer in the text. 

Line 50

Line 56

The manuscript in this section has been rephrased to make it clear that these efforts were made in an attempt to protect peatlands:

“In recent years the Indonesian peatland restoration agency “Badan Restorasi Gambut” (BRG) and NGO’s have attempted to initiate joint peatland restoration projects with communities in an effort to safeguard the local environment.”

6

The reasons for lack of community support are outlined clearly using subheadings and references are provided to substantiate claims, however more explicit use of examples or cross-reference to the context of the MRP might help connect section 3.1.1 to the initial aim of the methodology.

Line 130

Line 278

This section has received major revision and has rephrased.

7

The authors claim in section 3.1.2 that lack of community input has exacerbated socio-cultural issues, however this is not substantiated by a reference. Further research in this section is needed to make the argument that superficial mechanisms for involvement in restoration activities is an issue identified by members of Indonesian communities. 

Line 137

Line 161

A review of historical peatland restoration projects has been added which supplies multiple references substantiating this claim, most importantly Januar et al. 2021 and Puspitaloka et al. 2021.

8

The three-stage model is explained clearly however the image quality is quite poor making it difficult to read.

Line 256

Line 527

The image has been replaced with a higher quality version. It seems that the quality appears lower when uploaded onto the journal platform. We will work with the editors to ensure a high quality version is displayed on the final submission.

9

The concept of moral legitimacy is not clearly outlined in any stage of the process despite being discussed at length in the previous section. Integration of this concept into the model will help connect the relevancy of the two sections.

Line 256

Line 626

The government’s reputation (or moral legitimacy) is largely encompassed in the “interaction” pillar of the model. This has now been clarified in the manuscript:

“Interaction is the government’s main resource to improve its reputation or moral legitimacy with communities”.

10

The model, as is, seems to work well as a general base, however another rendition of the model with examples of the Indonesian context embedded in the diagram would make its relevance more immediately clear.

Line 477

An extensive background on past initiatives and the Indonesian use case for the model has since been added.

11

The section on involvement seems to miss a key aspect of ensuring effective co-management and deliberation which is identifying the needs of the community before imposing the needs of the business/government. The tone of this section seems to come from a top-down rather than a bottom-up perspective. (…). There is a strong focus on the need for intellectual involvement and education of the community in this process, however not every community member may have the capacity for this type of engagement and yet their values, wants, and needs must be considered. Perhaps more research is needed in this section to demonstrate that the authors understand that community members' values and needs must also (and arguable first) be considered to ensure mutual understanding.

Line 284

Line 605

Line 644

The “involvement” pillar was designed precisely to communicate governmental and national aims of peatland restoration to communities. This pillar is balanced by the “intervention” pillar, which represents the bottom-up flow of information, communicating local challenges among communities. The model therefore incorporates both perspectives and links them through the “interaction” pillar, to ensure an equal standing between both parties and adequate information exchange. This has been clarified in the manuscript as follows:

“The involvement pillar represents one side of a relationship which gives both parties an equal standing. It requires the government to involve communities in planning and decision-making of a restoration project. To make this possible, it builds on preliminary education to give communities a role above just providing labour.”

Intervention

“The intervention pillar represents the counterpart to involvement by giving communities a space to communicate their challenges in generating income and managing their land sustainably. This makes it possible to jointly develop ways of overcoming these barriers as a foundation of successful community involvement. Importantly, the intervention phase must be initiated by the government while giving communities full freedom to voice concerns and suggest solutions, which is enabled in the involvement phase.”

12

Similarly, the section on interaction seems to place more emphasis on the aspect of information dissemination from companies/governments rather than the process of aiming to understand the daily struggles and unique circumstances affecting community members.

Line 302

Line 644

This comment is addressed in the response above.

13

In the section on intervention, it is not immediately clear who should initiate the interventions or how these statements apply in the Indonesian context. Examples of methods to discuss and implement interventions in the Indonesian context would be helpful.

Line 318

Line 657

This comment is partially addressed in the response above. Additional examples have been added in the manuscript:

“This could be achieved through regular open forums with communities and government representatives, held consistently before project invitation until monitoring after completion. Concerns could also be anonymously voiced through key community representatives. Intervention might also involve negotiation on more formalized land rights which enable communities to pursue more long-term livelihood options.”

14

Success indicators in the last stage could include co-development of appropriate strategies rather than simply community approval and awareness.

Line 336

Line 686

As the goal of the process is to develop a social license, community approval is the ultimate goal and therefore a key variable to monitor. Co-development of response strategies has been explained in more detail in the manuscript:

“Additionally, they may inform necessary changes to livelihood alternatives, available resources or strategies to address any identified shortcomings within the project”

15

While this paper introduces a novel concept and provides some legitimate examples for how the new model applies to the Indonesian context there are some clear limitations which should be outlined in the manuscript. Some of these include 1.) whether the Indonesian government would elect to implement a strategy like this given the historical lack of consideration for local community involvement, 2.) the difficulty in enforcement capacity of this model due to historic and highly engrained power dynamic inequalities, and 3.) lack of clarity regarding the amount of time the process of obtaining an SLO would take and the conditions that must be considered for a successful start to this process.

Line 373

Line 742

Limitation 1 in our opinion does not require specific mentioning as we have since provided a detailed review of past restoration initiatives, indicating that the Indonesian government is making active attempts to involve communities in restoration. It is merely the method that needs to be improved.

We recognize 2 & 3 as potential limitations and have now added a discussion of them in the conclusion:

“Implementing our proposed model may be impacted by residual inequalities in the power dynamics within land management in Indonesia. The continuous nature of the SLR process also requires a level of stability which might be impacted by changes in government in the long-term. Increasing authority and autonomy will need to be shifted to communities early on to enable self-management in the long-term. Similarly, creating communication networks between communities will give them more leverage and enable mutual learning, shared resources and long-term cooperation.”

16

This model also does not account for understanding and preventing illegal resource extraction activities. Considering the aims of the paper, the authors do not adequately explain how this model could enable communities to actively restore peatlands, for example highlighting future research to assess the implementation of the model in restoration efforts. Careful consideration of further limitations of this model can help inform future research and allow others to attempt to implement this model.

Line 348

Line 745

A section has been added to the manuscript explaining how the model could be tested within the existing framework of the BRG restoration plan:

“Future research might aim to trial the SLR model using short- to medium-term success in-dicators and linking them through overarching long-term goals. Comparative studies be-tween communities implementing SLR and conventional methods could shed light on socio-economic and biophysical benefits of the concept.”

Specific Comments

17

Consider shortening or rewording this sentence to read more clearly as it is currently a run on.

Line 60-63

This comment has been addressed in the response to comment 1.

18

Can you define the three major area categories?

Line 101

Line 129

The land use types have now been defined.

19

Using the term degraded implies negative consequences for biodiversity and ecology, but not necessarily for rice production. It seems as though the communities and national government were motivated to clear more lands and restore degraded peatlands for continued economic production, perhaps this can be made clearer.

Line 112

This section has received major revision, with this sentence no longer being present.

20

"A lack of community input has exacerbated socio-cultural issues." A reference to the relevance of this statement in the Indonesian context would help qualify it.

Line 145

This sentence has been removed from the manuscript to make this section clearer.

21

Socio-political legitimacy is identified as a form of legitimacy however the figure shows socio-cultural. Perhaps these two should be aligned.

Line 195-196

Line 430

The figure and text have been aligned to both read “socio-cultural”

Reviewer 4 Report

An excellent paper and highly relevant - and raising important issues.

Author Response

We thank you for the wonderful feedback!

Reviewer 5 Report

In this article the authors apply the concept of the Social License to Operate (SLO) to peatland restoration in Indonesia, proposing a conceptual model for a Social License to Restore (SLR).

For two reasons, I do not find this article acceptable for publication in its current form. The first is inadequate information, and the second is a disconnect between concept and case study.

Inadequate Information

The Introduction and Historical Context sections give a cursory background on Indonesian peatlands but does not adequately familiarize the reader with Kalimantan. I suggest that a map and timeline be included, along with a more detailed description of the socioecological and socioeconomic complexities of these ecosystems and their human residents. For example, little information is provided on the MRP, and no information on the human population density, land ownership, hydrologic regimes, or extent of existing peatlands in the region. Neither have existing peatland restoration projects been given an adequate review. How many peatland restoration projects have been undertaken, and where? Have all of them failed, and all for the same reasons? How has this been documented?

How has SLO played out in Kalimantan? It seems to me that many business ventures around the world have taken place in the absence of SLO, and yet still been “successful” in some sense of the term. Has the SLO model been shown to work well in Indonesia? In general, connect this concept more clearly to business operations in your study site.

Also, it seems that the proposed SLR concept is similar to the long-existing concepts of Socioecological Systems (SES), Coupled Human and Natural Systems (CHANS), and participatory governance. These deserve some review in the present article.  

In summary, the Introduction could do much more to familiarize the reader with the conceptual and practical context of this situation.

Disconnect between concept and case

The basic premise of this article is: a) the current practice of peatland restoration in Kalimantan is failing; b) in the business world, the SLO model has been used to promote mutually beneficial activity; c) the SLO model can be applied to ecological restoration; and d) restoration under a social license model would be more successful.

I believe that a) and b) can be better explained with a more robust Introduction, as I have noted above. Items c) and d) are in the Results and Conclusions section of the article. I suggest that these sections can be better structured to provide a more compelling argument.

Regarding item c): It is not clear to me that business operation and ecological restoration necessarily follow the same “social license” model. The authors need to be clearer on why they see these two activities in parallel. For example, while business ventures typically involve the generation of profit, which may or may not be equitably distributed among the stakeholders. Some restoration activities, by contrast, generate no profit at all, and may in fact present a hindrance to some stakeholders. As a case in point, carbon sequestration in peatlands may well serve a global need but may be counterproductive to local needs.   

The authors can make a better case for the application of SLO to restoration by reviewing some case studies in which a social license approach has resulted in a successful and mutually beneficial restoration. An article that detailed several such cases from around the world would present a much stronger argument.  

Regarding item d): How can the authors be sure that such an approach would result in more successful restoration in Kalimantan? Lines 242-243 appear to make this conclusion. The evidence by which the authors arrive at this conclusion has not been made clear in the article. For example, after reading the article I found myself wondering if climate change, regional hydrologic regimes, or overarching demographic or economic factors could preclude successful peatland restoration here, no matter how much communication is achieved.   

The authors could demonstrate the efficacy of their concept through a well-documented case study in which participatory governance has been used in a successful restoration effort in Indonesia. In the absence of such evidence, the authors’ case is not well justified.   

Other notes

Shouldn’t the success indicators Ln 336 include measures of peatland restoration? 

The transition to SLO concepts on line 159 is abrupt; the authors might consider transitional language here.

Line 241 The Introduction of the SLR concept is also abrupt and could be better connected with the previous section.

Author Response

We have been delighted by the overall positive review response and thank you for your extensive feedback. The manuscript has received major revisions and improved significantly as a result. Please note that upon accepting all changes to view the manuscript properly, you might experience some slight formatting issues which can be corrected if/when the manuscript is accepted. Please find responses to your comments in the table below:

No.

Comment/Recommendation

Old position

New Position

Response

1

I suggest that a map and timeline be included, along with a more detailed description of the socioecological and socioeconomic complexities of these ecosystems and their human residents. For example, little information is provided on the MRP, and no information on the human population density, land ownership, hydrologic regimes, or extent of existing peatlands in the region. Neither have existing peatland restoration projects been given an adequate review. How many peatland restoration projects have been undertaken, and where? Have all of them failed, and all for the same reasons? How has this been documented?

Line 25-68

Line 121

The manuscripts historical context has been revised. The background now contains a review of the mega rice project, consequent land-reform and restoration projects to date (including a timeline). A contextual analysis of present-day central Kalimantan has also been added, reviewing hydrological regimes, the population demographics and land ownership.

A map has not been added as we believe it exceeds the scope of this article, which offers a theoretical perspective on peatland restoration.

The consequent discussion of our model and the conclusions have been adjusted according to the additional findings.

2

How has SLO played out in Kalimantan? It seems to me that many business ventures around the world have taken place in the absence of SLO, and yet still been “successful” in some sense of the term. Has the SLO model been shown to work well in Indonesia? In general, connect this concept more clearly to business operations in your study site.

Line 159

Line 477

SLO provides a structured avenue for businesses to ensure communities are involved with their activities and support them. Not striving to obtain a social license might still result in success but in this case the outcome is left to chance. This distinction has been clarified and a short review of the application of SLO in Indonesia has been included in the manuscript.

3

Also, it seems that the proposed SLR concept is similar to the long-existing concepts of Socioecological Systems (SES), Coupled Human and Natural Systems (CHANS), and participatory governance. These deserve some review in the present article.  

Line 159-178

Line 698

A short review of these concepts has been included under

3.5.4 The Novelty of SLR

4

The basic premise of this article is: a) the current practice of peatland restoration in Kalimantan is failing; b) in the business world, the SLO model has been used to promote mutually beneficial activity; c) the SLO model can be applied to ecological restoration; and d) restoration under a social license model would be more successful.

I believe that a) and b) can be better explained with a more robust Introduction, as I have noted above. Items c) and d) are in the Results and Conclusions section of the article. I suggest that these sections can be better structured to provide a more compelling argument.

Line 130-158 and Line 159-178

Line 389

The structure has been revised as follows:

“3.5 Social License to Operate Concepts

3.5.1 Components of a Social License

3.5.2 Applying SLO in Indonesian government-led restoration

3.5.3 The Social License to Restore

3.5.4 The Novelty of SLR

4 Conclusions”

5

Regarding item c): It is not clear to me that business operation and ecological restoration necessarily follow the same “social license” model. The authors need to be clearer on why they see these two activities in parallel. For example, while business ventures typically involve the generation of profit, which may or may not be equitably distributed among the stakeholders. Some restoration activities, by contrast, generate no profit at all, and may in fact present a hindrance to some stakeholders. As a case in point, carbon sequestration in peatlands may well serve a global need but may be counterproductive to local needs.   

Line 241-255

Line 477

The manuscript has been amended to clarify the similarities and differences un applying SLO in a restoration vs business concept under

3.5.2 Applying SLO in Indonesian government-led restoration.

6

The authors can make a better case for the application of SLO to restoration by reviewing some case studies in which a social license approach has resulted in a successful and mutually beneficial restoration. An article that detailed several such cases from around the world would present a much stronger argument. 

Line 161

Line 477

The application of SLO in the public restoration space is a novel approach which has not previously been attempted. We believe that the added review of restoration initiatives in Indonesia earlier in the paper as well as the review of past applications of SLO in Indonesia provides a reliable basis for assessing the potential outcome of applying SLO in ecological restoration.

7

Regarding item d): How can the authors be sure that such an approach would result in more successful restoration in Kalimantan? Lines 242-243 appear to make this conclusion. The evidence by which the authors arrive at this conclusion has not been made clear in the article. For example, after reading the article I found myself wondering if climate change, regional hydrologic regimes, or overarching demographic or economic factors could preclude successful peatland restoration here, no matter how much communication is achieved

Line 242-243

Line 717

After having provided a more detailed background on the target regions demographics, hydrological regimes and economic context the results and conclusion have been revised.

8

The authors could demonstrate the efficacy of their concept through a well-documented case study in which participatory governance has been used in a successful restoration effort in Indonesia. In the absence of such evidence, the authors’ case is not well justified.   

Line 161

The documented historical context and restoration projects to date indicate that previous restoration efforts in Indonesia might have failed precisely due to a lack of adequate participatory governance. The added background indicates that participatory governance has successfully highlighted challenges relating to community acceptance, but a more rigorous engagement process is needed to actually resolve them.

9

Shouldn’t the success indicators Ln 336 include measures of peatland restoration? 

Line 690

The framework focuses primarily on enabling community participation in restoration, which precedes actual ecosystem rehabilitation. However, the link of community approval to biophysical restoration indicators has been clarified.

10

The transition to SLO concepts on line 159 is abrupt; the authors might consider transitional language here.

Line 389

The beginning of “SLO Concepts” has been revised to read as follows:

“The three overarching issues communities face in rural Kalimantan cam be argued to preclude the Indonesian government from gaining their approval. In business literature such an approval has often been termed a “social license to operate”.”

11

Line 241 The Introduction of the SLR concept is also abrupt and could be better connected with the previous section.

Line 517

This section has been revised entirely to fit better into sections before and after. The section now leads with a justification on how SLO is applicable in a restoration scenario and then transitions into the actual model design as follows:

“In light of these reasons, we argue that SLO can find use in a government-led restoration initiative if its components are adapted and implemented as actions rather than goals. The SLR model involves three stages, firstly creating a mutual level of understanding, then engaging the community through involvement, interaction and intervention and finally evaluating progress through relevant performance indicators (Figure 3).”